

# Bioinformatic and biochemical analysis of the key binding sites of the pheromone binding protein of *Cyrtotrachelus buqueti* Guerin-Meneville (Coleoptera: Curculionidea)

Hua Yang[1,*], Yan-Lin Liu[1,*], Yuan-Yuan Tao[1], Wei Yang[1], Chun-Ping Yang[1], Jing Zhang[2], Li-Zhi Qian[1], Hao Liu[1] and Zhi-Yong Wang[3]

[1] Sichuan Agricultural University, Key Laboratory of Ecological Forestry Engineering of Sichuan Province/ College of Forestry, Chengdu, Sichuan, China
[2] Provincial Key Laboratory of Agricultural Environmental Engineering, Sichuan Agricultural University, Chengdu, China
[3] Key Laboratory of Control and Resource Development of Bamboo Pest of Sichuan Province, Leshan, China
[*] These authors contributed equally to this work.

## ABSTRACT

The bamboo snout beetle *Cyrtotrachelus buqueti* is a widely distributed wood-boring pest found in China, and its larvae cause significant economic losses because this beetle targets a wide range of host plants. A potential pest management measure of this beetle involves regulating olfactory chemoreceptors. In the process of olfactory recognition, pheromone-binding proteins (PBPs) play an important role. Homology modeling and molecular docking were conducted in this study for the interaction between CbuqPBP1 and dibutyl phthalate to better understand the relationship between PBP structures and their ligands. Site-directed mutagenesis and binding experiments were combined to identify the binding sites of CbuqPBP1 and to explore its ligand-binding mechanism. The 3D structural model of CbuqPBP1 has six a-helices. Five of these a-helices adopt an antiparallel arrangement to form an internal ligand-binding pocket. When docking dibutyl phthalate within the active site of CbuqPBP1, a CH-$\pi$ interaction between the benzene ring of dibutyl phthalate and Phe69 was observed, and a weak hydrogen bond formed between the ester carbonyl oxygen and His53. Thus, Phe69 and His53 are predicted to be important residues of CbuqPBP1 involved in ligand recognition. Site-directed mutagenesis and fluorescence assays with a His53Ala CbuqPBP1 mutant showed no affinity toward ligands. Mutation of Phe69 only affected binding of CbuqPBP1 to cedar camphor. Thus, His53 (Between α2 and α3) of CbuqPBP1 appears to be a key binding site residue, and Phe69 (Located at α3) is a very important binding site for particular ligand interactions.

Corresponding author
Wei Yang, ywei0218@aliyun.com

## INTRODUCTION

During long-term evolution insects have developed a sensitive sense of smell, which enables insects to detect external volatile semiochemicals when searching for various environmental cues, such as foraging for food, finding a breeding partner and locating a spawning ground (*Gu et al., 2011*; *Larsson et al., 2004*). Tentacles are the main olfactory part of insects and contain a large variety of receptors. Receptors are widely distributed with various olfactory-rated functional proteins, including odorant binding proteins (OBPs), chemosensory proteins (CSPs) and olfactory receptors (Ors). OBPs are divided into pheromone binding proteins (PBPs), general odorant binding proteins (GOBPs) and antennal binding proteins (ABPs) (*Vogt & Riddiford, 1981*). Research on the binding mechanism between OBPs and ligand molecules has been a major focus of research, including defining the three-dimensional (3D) structure of these OBPs. *Kruse et al. (2003)* and *Thode et al. (2008)* initially analyzed the general odorant binding protein (LUSH) of *Drosophila melanogaster* and the crystal structure of the complex between LUSH and alcohol, and clarified that Thr57 is a key residue involved in ligand interaction. In accordance with X-ray diffraction analysis of the pheromone binding protein BmorPBP of *Bombyx mori* and structure of bombykol, *Sandler et al. (2000)* discovered that Ser56 of this protein played a key role by forming a hydrogen bond with the ligand bombykol. According to the structures of odorant binding protein CquiOBP1 and MOP of *Culex quinquefasciatus*, *Mao et al. (2010)* discovered that instead of hydrogen bonds, the interaction between protein and ligand was driven by van der Waals forces and hydrophobic interactions. Based on the structure between the odorant binding protein HoblOBP2 of *Holotrichia oblita* and ethyl benzenecarboxylate, *Zhuang et al. (2013)* discovered that this protein-ligand complex involved both van der Waals forces and hydrophobic interactions. Currently, high-resolution structural data describing the complex between the pheromone binding protein of *Cyrtotrachelus buqueti* and an odor molecule is unavailable, and thus information about the mode of action of this protein remains unresolved.

*Cyrtotrachelus buqueti* (*C. buqueti*) also named as the bamboo snout beetle, belongs to Cyrtotrachelus, Curculionidea, Coleoptera. *C. buqueti* is endangering survival of bamboo shoots from 28 different types of bamboos, including *Bambusa*, *Dendrocalamopsis* and *Dendrocalamus*. In particular, the larvae prefer the bamboo shoots of *Phyllostachys pubescens*, *Dendrocalamopsis oldhami*, *Bambusa textilis*, *Bambusa pervariabilis*, *Dendrocalamopsis daii* and other sympodial bamboo species (*Ju et al., 2005*; *Wang, Chen & Wang, 2005*). *C. buqueti* is distributed widely in the Sichuan Province, Chongqing City, Guangdong Province, Guangxi Province, Guizhou Province and other provinces (districts) as well as Vietnam, Burma, Thailand and other countries and regions in Southeast Asia (*Yang et al., 2009*). *C. buqueti* is one of 233 hazardous forest pests issued in 2003 for the first time (*Yang et al., 2015*).

Currently, research on *C. buqueti* has mainly concentrated on a description of the general biological characteristics and common chemical pest control approaches (*Ju et al., 2005*; *Wang, Chen & Wang, 2005*; *Yang et al., 2010*; *Yang et al., 2009*). The development of sex attractants remains poorly understood. *Mang et al. (2012)* have extracted and studied
the body surface semiochemicals of *C. buqueti* adults, whereas *Yang et al. (2017a)* have constructed a transcriptome library of *C. buqueti* and analyzed the sex pheromone binding protein gene. *Yang et al. (2017b)* have also cloned the sex pheromone binding protein gene that codes for the protein CbuqPBP1, and conducted fluorescence competitive binding assays for many types of simple odor substances. Based on a phylogenetic analysis (*Yang et al., 2018*; summarized in supplementary information) CbuqPBP1 was quite similar to the PBPs of other insects. Amino acid sequence similarity analysis showed that CbuqPBP1 had 37.68% similarity with 27 PBPs of 17 insects of Coleoptera and Lepidoptera. The similarities with PBPs from Coleoptera and Lepidoptera were 38.47% and 52.39% respectively.

In this paper, homology modeling of the pheromone binding protein CbuqPBP1 of *C. buqueti* has been conducted to create a 3D model of the protein. Molecular docking has also been carried out to define the interaction mode between the ligand dibutyl phthalate and CbuqPBP1. Two key binding site residues, Phe69 and His53, were identified from this modeling and were mutated. Fluorescence competitive binding assays were conducted for these mutants and binding mechanism between CbuqPBP1 and odor molecules was analyzed. The results provide a platform for using pheromones to prevent and control *C. buqueti* efficiently.

## MATERIALS & METHODS

### Materials

Three compounds were chosen to investigate the ligand-binding specificity of CbuqPBP1. Ligands of the highest purity were purchased from Aladdin (Shanghai, China) and stored in accordance with the manufacture's specifications. The sequence of CbuqPBP1 was taken from the GenBank with accession number KU845733.1.

### Alignment and homology modeling

The amino acid sequence of CbuqPBP1 was downloaded from the GenBank and Blast was used to search against the CbuqPBP1 protein sequence in the Protein Data Bank to identify a structural template. Software Modeller 9.19 (http://salilab.org/modeller/) was used for homology modeling based on the sequence comparison results with the structural template sequence identified. The 3D structure obtained from modeling was evaluated with SAVES v5.0 (https://servicesn.mbi.ucla.edu/SAVES/). After confirming the models, the Chiron on-line server was used for optimization. Modeller 9.19 was used to optimize loop regions and PyMOL was used to analyze structural characteristics and to search for ligand binding sites.

### Molecular docking

Based on the established homology model, the docking program AUTODOCK vina 1.1.2 was used to find the potential binding mode between CbuqPBP1 and the ligand dibutyl phthalate. Dibutyl phthalate with strong affinity is a female pheromone of the giant bamboo weevil, which plays a role in the process of male individual searching for female individual. ChemBioDraw Ultra 14.0 was used to simulate the structure of dibutyl phthalate and to generate a 3D structure of the ligand. Energy optimization was conducted using the

MMFF94 force field and Autodock Tools 1.5.6 was used to create the PDBQT format (*Huey et al., 2007*; *Morris et al., 2009*). Binding coordinates of CbuqPBP1 and dibutyl phthalate were set to: center_x = 22.389, center_y = −25.143, center_z = 1.08, and size_x = 15, size_y = 15, size_z = 15. Parameter exhaustiveness was set to 20 and default values were used for other parameters to increase the calculation accuracy. Finally, the conformation with the highest score was selected and PyMoL 1.7.6 was used for visual inspection and analysis of the structural data.

## Site-directed mutagenesis

The CbuqPBP1 coding sequence was mutated to yield the two mutants CbuqPBP1-Phe69A (phenylalanine to alanine at position 69) and CbuqPBP1-His53A (histidine to alanine at position 53). PCR reactions were used to form overlapping chains. The extension of overlapping chains was used to splice segments in a superimposed manner. Primer5 was used to design primers (Table 1). Three rounds of PCR amplification were conducted after designing primers. Expression vectors (pET-28a(+)/PBP1-Phe69A, pET-28a(+)/PBP1-His53A and pET-28a(+)/PBP1) were generated and transformed into *Escherichia coli* BL21(DE3) competent cells for protein overexpression. Recombinant proteins produced were detected by SDS-PAGE analysis.

## Expression and purification of the native protein and mutants

Expression plasmids were transformed into *E. coli* TOP10 competent cell and plated on agar plates. Several colonies were selected randomly for overnight cultivation in LB media and plasmids were extracted for sequencing. Mutant plasmids pET-28a(+)/PBP1- Phe69A and pET-28a(+)/PBP1- His53A with the correct sequence were transformed into *E. coli* BL21(DE3) competent cells, and cells were grown to an optical density at 600 nm ($OD_{600}$) of 0.6. IPTG was added to the culture to a final concentration of 1 mM and cells were further grown at 37 °C with shaking for 3 h to induce protein expression (*Deng et al., 2011*). After harvesting cells by centrifugation, ultrasound sonication was used to disrupt cells (200 W, 3/4 s, 25–30 min). The supernatants and sediments were collected under low temperature centrifugation (16, 000 g-force, 50 min) and SDS-PAGE detection was conducted. Nickel affinity (Ni-NTA) was used to purify recombination proteins, and the purified proteins were stored in Tris-HCl buffer (pH 7.4, 50 mM). To avoid the function of the protein being affected by the His-tag, recombinant bovine enterokinase was used to remove the His-tag and the protein was re-purified and collected. Purity was confirmed by SDS-PAGE analysis.

## Fluorescence assay

To measure the affinity of the fluorescent ligand N-phenyl-1-naphthylamine (1-NPN) toward CbuqPBP1, a 2 μM solution of protein in 50 mM Tris-HCl, pH 7.4, was titrated with aliquots of 1 mM 1-NPN dissolved in methanol to a final concentration of 16 μM. The probe was excited at 337 nm and emission spectra were recorded between 350 and 550 nm. To evaluate the effect of pH on the binding affinity of CbuqPBP1, we also measured its binding with 1-NPN over a pH range of 4.5–9.0. The displacement of 1-NPN by selected ligands was measured in a competitive binding assay using both the protein

**Table 1   Mutagenic primers for CbuqPBP1.**

| Primers | Sequence |
|---|---|
| PBP1-F69A-Fm | 5′-aatgcactattttctgtacagcgaaaaaattcgatttgatgaaag-3′ |
| PBP1-F69A-Rm | 5′-ctttcatcaaatcgaatttttcgctgtacagaaaatagtgcatt-3′ |
| PBP1-H53A-Fm | 5′-gatatccaagctctgatgaacgcggaacgaccagtcacccatgc-3′ |
| PBP1-H53A-Rm | 5′-gcatgggtgactggtcgttccgcgttcatcagagcttggatatc-3′ |
| PBP1-F | 5′-ggaattccatatgcttagcgaaagcttagttgttgatg-3′ |
| PBP1-R | 5′-ccgctcgagttaaaaactggtaattccaag-3′ |

and 1-NPN at 2 μM. The mixtures were titrated with 1 mM methanol solutions of each competitor at concentrations of 2–16 μM. The fluorescence of the mixture was recorded after 5 min. Dissociation constants for 1-NPN and the stoichiometry of binding were obtained from Scatchard plots of the binding data using the Prism software. For other competitor ligands, the dissociation constants were calculated from the corresponding half maximal inhibitory concentration ($IC_{50}$) values using the equation: inhibitory constant Ki = $[IC_{50}]/(1+[1\text{-NPN}]/K_{1\text{-NPN}})$, where [1-NPN] is the free concentration of 1-NPN and $K_{1\text{-NPN}}$ is the dissociation constant of the protein/1-NPN complex.

## RESULTS

### Three-dimensional model of CbuqPBP1

On the basis of the Blast search against the Protein Data Bank, two types of insect odor proteins with known structures and quite similar sequences to the CbuqPBP1 sequence were found. These two odorant binding proteins were *Nasonovia ribisnigri* OBP3 (NribOBP3 PDB ID: 4Z45_A) and *Megoura viciae* OBP3 (MvicOBP3 PDB ID: 4Z39_A). The total sequence identity between the target (CbuqPBP1) and the template protein (NribOBP3) is 33% (*Cavasotto & Phatak, 2009*) (Fig. 1A). The resolution of the template is 2.02 Å.

After homology modeling, the 3D structure of CbuqPBP1 (Fig. 1B) is clearly very similar to the 3D structure of the template NribOBP3 (Fig. 1C). The structural characteristics of CbuqPBP1 are similar to other sex pheromone binding proteins and include six α-helices: residues 26–36 (α1), 44–51 (α2), 59–72 (α3), 83–94 (α4), 101–114 (α5) and 123–137 (α6). Six conserved cysteine residues stabilize the protein structure by forming three disulfide bonds. Disulfide bond Cys36–Cys67 connects α1 and α3, Cys63–Cys121 connects α3 and α6, and Cys110–Cys130 connects α5 and α6. Five of the α-helices adopt an antiparallel arrangement (α1, α3, α4, α5 and α6) and form an internal binding pocket. α2 forms a cover-type structure or lid above the pocket, which stabilizes this structure.

The result of further rationality estimates by Pro-CHECK (Fig. 1D) was that 88.4% residues were in the favored regions (red area A, B and L), 10.1% of the residues fall into additionally allowed regions (bright yellow area a, b, l, p) and 0.8% residues have backbone torsion angles that fall into generously allowed regions (light yellow area ~a, ~b, ~l, ~p). The percentage sum of residues in the allowed regions was 99.3%, which was higher than 95%. This result showed that the constructed 3D structure of CbuqPBP1 was a high-quality model.

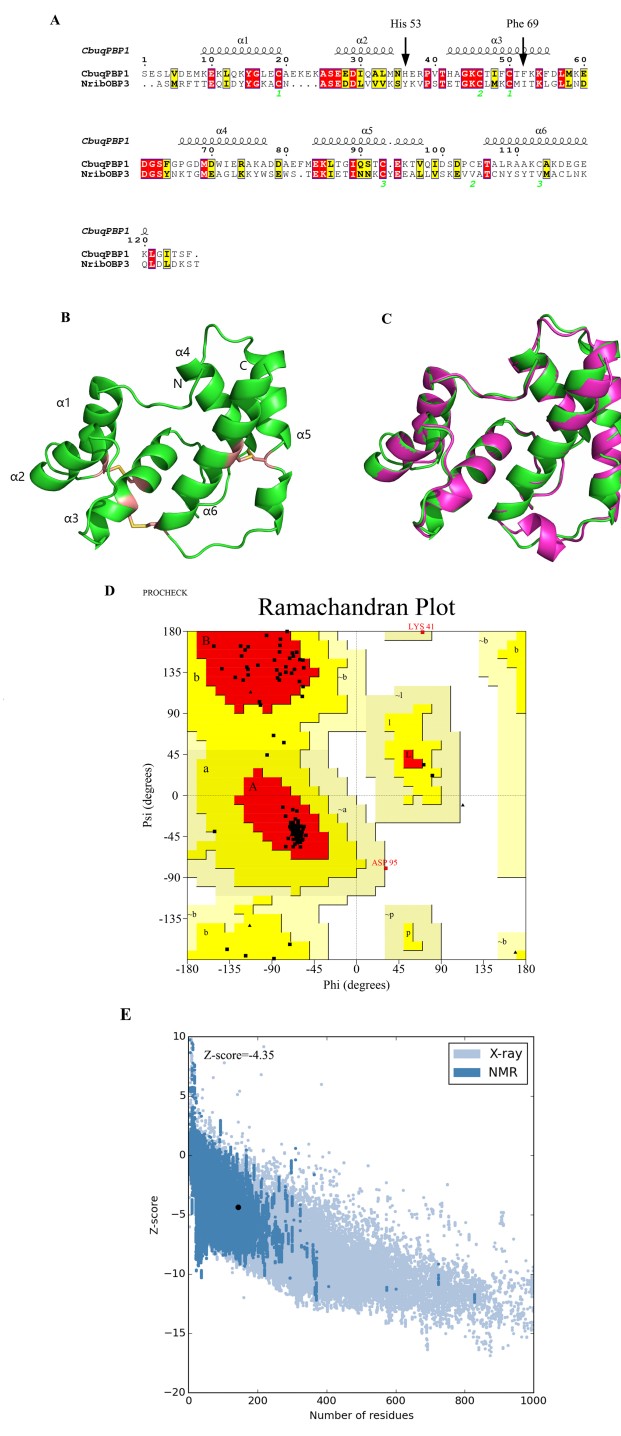

**Figure 1  Three-dimensional (3D) model of CbuqPBP1.** (A) Sequence alignment between CbuqPBP1 and NribOBP3. (B) 3D structure of CbuqPBP1. The N and C termini and the six a-helices are labeled and the three disulfide linkages are shown in yellow stick representations. (C) Superimposed penetrative structure of CbuqPBP1 and NribOBP3. The model of CbuqPBP1 and crystal structure of NribOBP3 are shown in green and violet, respectively. (D) The results of the PROCHECK evaluation of the CbuqPBP1 model. (E) Overall model quality.

Energy assessment was performed on ProSa Fig. 1E. The shadow part is Z-score value of all proteins similar to Cbuq PBP1 protein in PDB database, and the black spot is Z-score value of Cbuq PBP1 protein, which is −4.35. The Z-score value of template protein NOBP3 is −5.87 in the range of Z-score of known reasonable structural proteins, which indicates that the modeling structure is more stable than template structure. This indicates that the homologous modeling Institute is more stable than template structure. The constructed CBuq PBP1 protein is reasonable in energy.

## Molecular docking

To research characteristics of CbuqPBP1 binding with odor molecules, dibutyl phthalate (Fig. 2A), which interacts with CbuqPBP1 favorably, was selected to construct a complex between CbuqPBP1 3D model and dibutyl phthalate. Such a model should clarify the mode of interaction of dibutyl phthalate with CbuqPBP1 at the molecular level. We have docked dibutyl phthalate with the active pocket of CbuqPBP1, with a binding energy of −6.4 kcal/mol. Generally, compound dibutyl phthalate bound to the active pocket of CbuqPBP1 with a compact conformation (Fig. 2B).

The benzene ring and one aliphatic chain of dibutyl phthalate were located in the hydrophobic region at the bottom of the pocket. Strong hydrophobic interactions formed between the ligand and residues Leu3, Leu4, Leu5, Leu29, Leu50, Pro56, Ile65 and Phe69. Another side chain of dibutyl phthalate was located at the opening of the pocket. Based on detailed analysis, a CH-$\pi$ interaction may occur between the benzene ring of dibutyl phthalate and residue Phe69. Moreover, an important long-range hydrogen bond (3.3 Å) can form between one ester carbonyl oxygen of dibutyl phthalate and residue His53 (Fig. 2C). All aforementioned interactions enable the formation of a stable complex between dibutyl phthalate and CbuqPBP1.

## Site-directed mutagenesis of CbuqPBP1 and binding specificities of mutants

After double enzyme digestion with restriction enzymes *Nde* I and *Xha* I, mutant plasmids pET-28a(+)/CbuqPBP1-His53A and pET-28a(+)/CbuqPBP1-Phe69A, and the original plasmid pET-28a(+)/PBP1 formed bands in an agarose gel that were ~400 bp in length (Fig. 3). After SDS-PAGE analysis of protein overexpression, three specific bands with molecular weights of 16 kDa were observed in the SDS-PAGE gel, which is consistent with expected molecular weight of the target proteins (Fig. 4).

After ultrasonication to disrupt the bacteria and release the recombinant proteins (including His tag), SDS-PAGE analysis could be conducted (Fig. 5). All recombinant proteins were found in the supernatant part of the disrupted cells. After purification, recombinant bovine enterokinase was used to cleave the His-tag and following a further round of purification pure recombinant protein samples were obtained.

1-NPN was selected as the fluorescent probe. Fluorescence competitive binding assays were conducted for the purified wild-type CbuqPBP1, mutant CbuqPBP1-His53A and CbuqPBP1-Phe69A proteins. The fluorescence peak maximum in the presence of the recombinant proteins was recorded at different concentrations. The Scatchard equation was

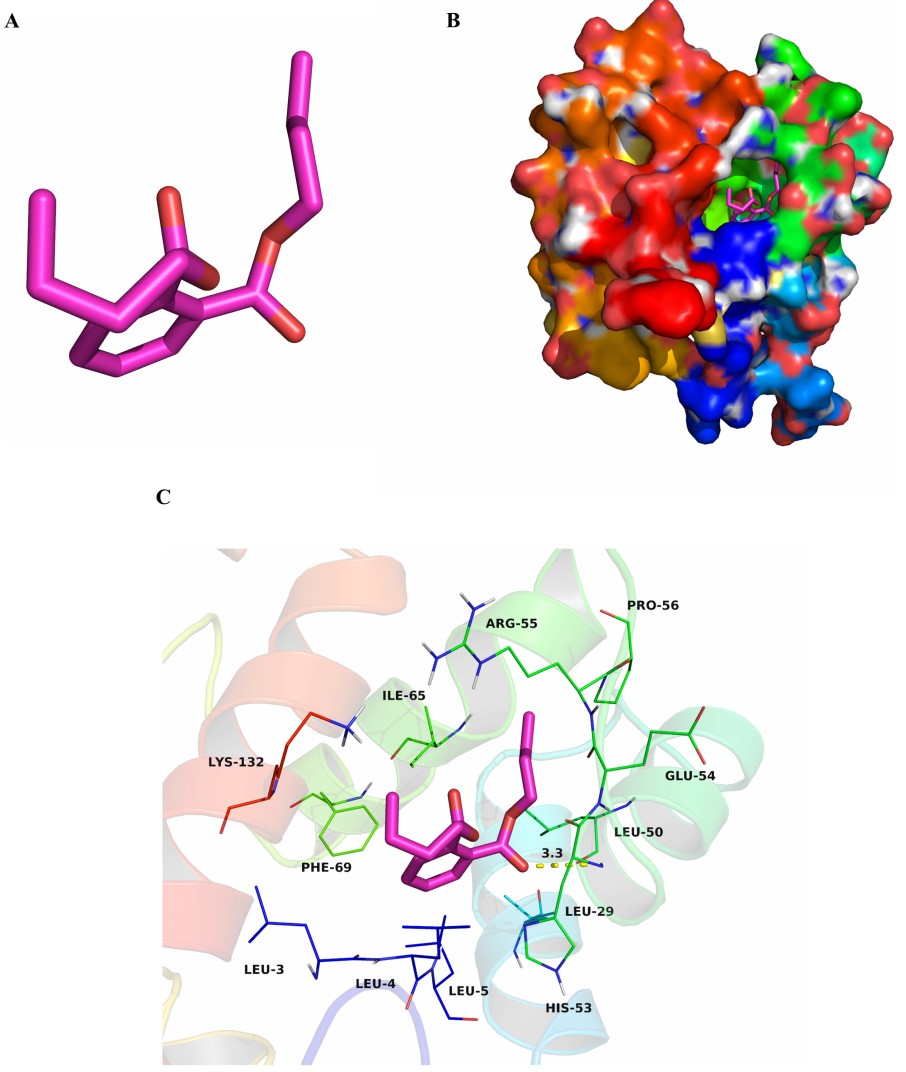

**Figure 2** **The binding pocket of CbuqPBP1 and the docking result with dibutyl phthalate.** (A) Tertiary structure of dibutyl phthalate. (B) The binding pocket of CbuqPBP1 and dibutyl phthalate docked into the active site of the CbuqPBP1 receptor. (C) Diagram of the van der Waals interactions and hydrophobic interactions of dibutyl phthalate with key binding site residues.

used to calculate the equilibrium binding constant ($K_d$) between CbuqPBP1, CbuqPBP1-His53A, CbuqPBP1-Phe69A and 1-NPN, which were determined to be 2.725, 3.352 and 2.260 µM, respectively. When the final concentration of odor substance was higher than 50 µM, the fluorescence peak did not decrease to half its value. This showed that almost no affinity was established between protein and the odor substance, and the binding constant could not be calculated (Fig. 6).

Dibutyl phthalate, benzothiazole and cedar camphor were selected based on previous fluorescence binding assay test (*Yang et al., 2017b*). Fluorescence competitive binding assays were conducted with CbuqPBP1, CbuqPBP1-His53A and CbuqPBP1-Phe69A (Fig. 7).

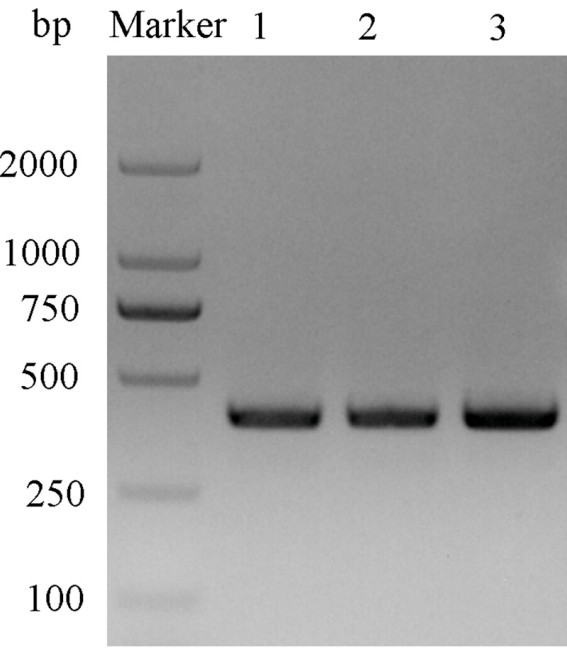

**Figure 3** **Double digestion map of the mutant and wild-type plasmids.** Lane Marker: protein molecular weight standard; Lane 1: pET-28a (+)/PBP1-Phe69A; Lane 2: pET-28a (+)/PBP1-His53A; and Lane 3: pET-28a (+)/PBP1.

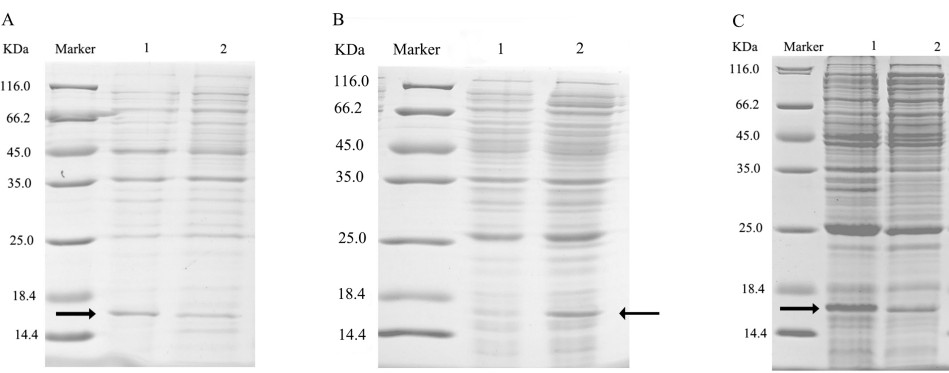

**Figure 4** **SDS-PAGE analysis of the total bacterial protein lysate of the mutant and wild-type CbuqPBP1.** (A) CbuqPBP1-His53A mutant. Lane 1: IPTG induced total protein lysate; Lane 2: total protein lysate without IPTG induction. (B) CbuqPBP1- Phe69A mutant. Lane 1: total protein lysate without IPTG induction; Lane 2: IPTG induced total protein lysate. (C) wild-type CbuqPBP1. Lane 1: IPTG induced total protein lysate; Lane 2: total protein lysate without IPTG induction.

Based on the results, CbuqPBP1 bound favorably with dibutyl phthalate, benzothiazole and cedar camphor. The binding ability of CbuqPBP1-His53A with the three types of odor substances was essentially lost. The binding ability of CbuqPBP1-Phe69A mutant with cedar camphor was significantly reduced, whereas affinity toward the other two odor substances was not significantly different from that of the wild-type protein (Table 2).

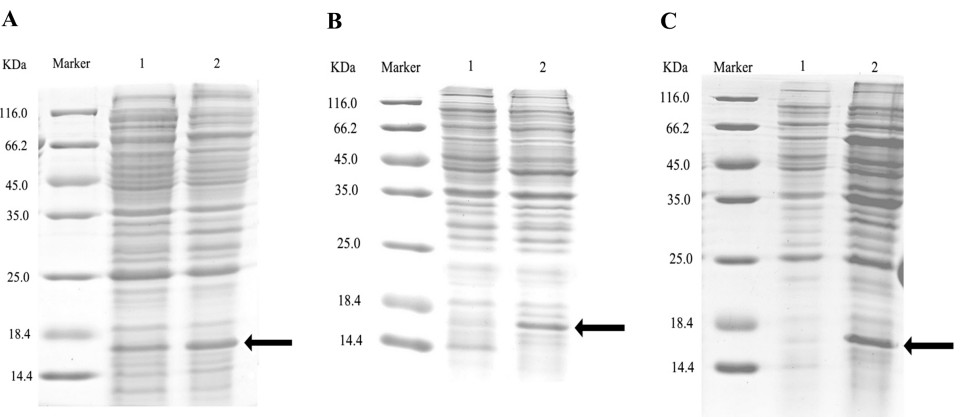

**Figure 5** **SDS-PAGE analysis of supernatant and precipitant of bacterial fragmentation following expression of the mutant and wild-type CbuqPBP1.** Lane 1: IPTG induced expression of insoluble material; Lane 2: IPTG induced expression of supernatant following cell disruption by sonication. (A) Wild-type protein. (B) CbuqPBP1-His53A mutant. (C) CbuqPBP1- Phe69A mutant.

## DISCUSSION

Currently, 3D structure prediction of odorant binding proteins through homology modeling has been conducted for proteins from *Choristoneura rosaceana, Choristoneura murinana, Pectinophora gossypiella, Heliothis assulta, Spodoptera exigua, Spodoptera exigua, Holotrichia oblita, lettuce Aphidoidea, Megoura viciae* and other insects (*Northey et al., 2016*; *Sun et al., 2013*; *Wang et al., 2015*). On the basis of homology modeling of pheromone binding protein CbuqPBP1 of *C. buqueti*, the 3D structure is composed of six α-helices, which packed together and were stabilized by three disulfide bonds. Disulfide bonds Cys36–Cys67, Cys63–Cys121 and Cys110–Cys130 connected α1 and α3, α3 and α6, α5 and α6 respectively. Five of the α-helices arranged in an antiparallel manner to form an internal binding pocket (*Tian et al., 2017*). α2 formed a cover-type structure above the pocket, which was similar to *Holotrichia oblita* HoblOBP2 (*Zhuang et al., 2013*) structures. As for 3D structure of *Bombyx mori* BmorPBP, four antiparallel α-helices formed a hydrophobic pocket and α2 and α3 did not participate in the formation of the pocket (*Sandler et al., 2000*). This might be due to differences in hydrophobic pocket of the 3D structure of odorant binding proteins from different insects. Such differences are likely to be closely related to the function of these proteins.

According to research, odorant binding proteins from some insects interact with their cognate ligand through hydrogen bonds and hydrophobic interactions, whereas other odorant binding proteins from other insects interact with odorants via van der Waals forces and hydrophobic interactions (*Sandler et al., 2000*). In this report, a CH-$\pi$ interaction formed between the benzene ring of dibutyl phthalate and Phe69. This CH-$\pi$ interaction is generally considered to be a relatively weak hydrogen bond. Previous research has indicated that CH-$\pi$ interactions are important in carbohydrate–protein identification processes, where the CH-$\pi$ features as a synergistic interaction that plays an important role in stabilizing the structure of the complex (*Jiang et al., 2009*; *Kozmon*

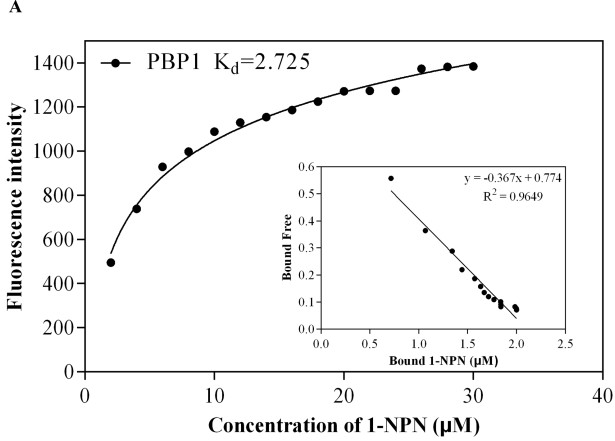

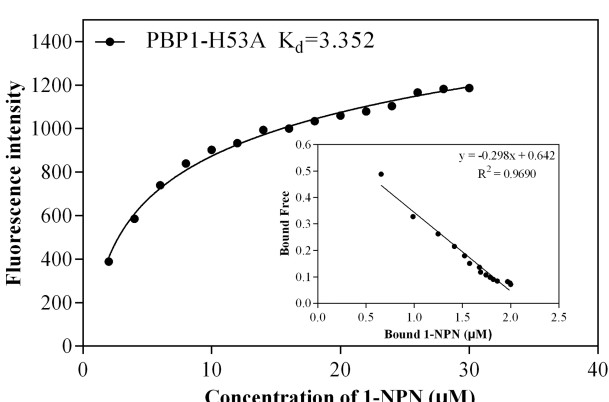

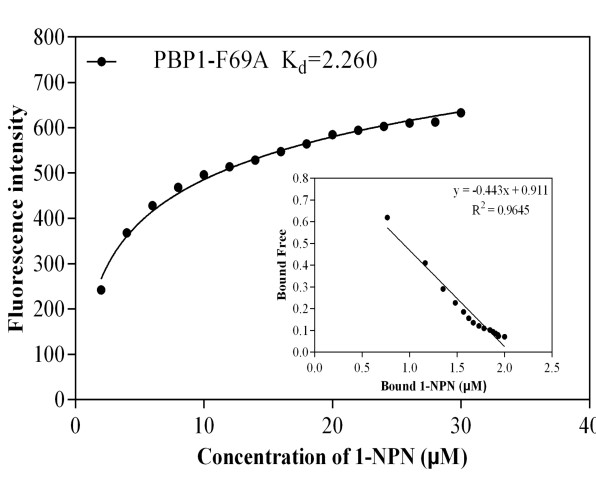

**Figure 6** **The binding curve and $K_d$ of mutant and wild-type CbuqPBP1 toward 1-NPN.** (A) Wild-type protein. (B) CbuqPBP1-His53A mutant. (C) CbuqPBP1- Phe69A mutant.

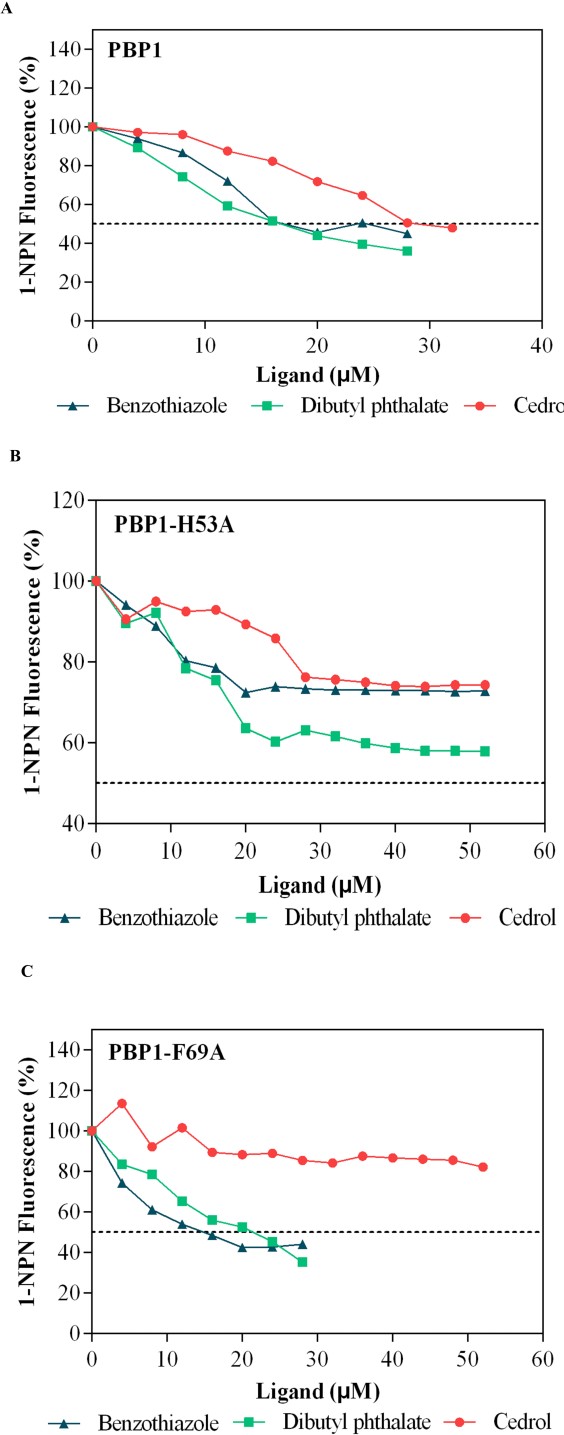

**Figure 7** **Competitive binding curves of selected ligands toward mutant and wild-type CbuqPBP1.** (A) Wild-type protein. (B) CbuqPBP1-His53A mutant. (C) CbuqPBP1- Phe69A mutant.

**Table 2  Binding ability of ligands to mutant and wild-type CbuqPBP1.**

| Ligands | IC50 (μM) | | | Ki (μM) | | |
|---|---|---|---|---|---|---|
| | PBP1 | H53A | F69A | PBP1 | H53A | F69A |
| benzothiazole | 13.426 | – | 10.538 | 9.822 | – | 7.305 |
| dibutyl phthalate | 16.889 | – | 20.04 | 12.355 | – | 13.893 |
| cedrol | 29.953 | – | – | 21.912 | – | – |

*et al., 2011*). The CH-$\pi$ interaction involves a nonpolar interaction between the CH proton and electron-rich aromatic ring $\pi$ electron cloud system, playing a similar role to hydrogen bonding in controlling crystal stacking, maintaining biomolecular structures and participating in molecular recognition processes (*Ye et al., 2015*; *Zhao et al., 2014*). Therefore, we hypothesize that the CH-$\pi$ interaction may play a role in binding and stabilizing the interaction with odor molecules.

An ester carbonyl oxygen from dibutyl phthalate and His53 from the protein formed a weak 3.3 Å hydrogen bond. Such a hydrogen bond has been reported in odorant binding proteins of other insects, for example, BmorPBP1 of *B. mori* and pheromone compound interacted through a hydrogen bond. General odorant binding protein (LUSH) from *Drosophila melanogaster* and the pheromone binding protein (ApolPBPl) from *Antheraea polyphemus* interact with their cognate ligands through hydrogen bonds (*Damberger et al., 2007*; *Thode et al., 2008*).

According to the fluorescence competitive binding assay, mutant pET-28a(+)/PBP1-His53A could not interact with odor substances. Replacing His53 with alanine removed the ability of the mutant to form this key hydrogen bond with ligands, and therefore the ability to bind with odor substances. Thus, His53 is a key binding site residue of the pheromone binding protein of *C. buqueti*. Mutein pET-28a(+)/PBP1-Phe69A did not bind cedar camphor. However, only a decrease in binding ability toward dibutyl phthalate and benzothiazole was observed. These observations indicate that only a small number of intermolecular forces between the protein and odor molecules were affected by this mutation (*Zhuang et al., 2014*). Thus, the binding affinity had been reduced, but not completely lost.

## CONCLUSIONS

In summary, we hypothesized that the CbuqPBP1 interaction and release of the ligand involves hydrogen bond formation via His53. Phe69 is the binding site for CbuqPBP1 to combine with odor substance; however, Phe69 is not a key binding site residue. Moreover, these observations showed that the combination between CbuqPBP1 and ligands was affected by loss of hydrogen bonding and other intermolecular forces, and the interaction between CbuqPBP1 and ligands involves the joint action of many acting forces and the binding site (*Li et al., 2016*).

## ACKNOWLEDGEMENTS

The authors thank Liwen Bianji, Edanz Editing China for editing the English text of a draft of this manuscript.

### Funding

This work was funded by the Key Fund of the Education Department in Sichuan (17ZB0344) and the Key Laboratory Fund for Scientific Research in Sichuan (003Z1401). The funders had no role in study design, data collection and analysis, decision to publish, or preparation of the manuscript.

### Grant Disclosures

The following grant information was disclosed by the authors:
Education Department in Sichuan: 17ZB0344.
Key Laboratory Fund for Scientific Research in Sichuan: 003Z1401.

### Competing Interests

The authors declare there are no competing interests.

### Author Contributions

- Hua Yang and Yan-Lin Liu conceived and designed the experiments, performed the experiments, analyzed the data, contributed reagents/materials/analysis tools, prepared figures and/or tables, authored or reviewed drafts of the paper, approved the final draft.
- Yuan-Yuan Tao conceived and designed the experiments, performed the experiments, contributed reagents/materials/analysis tools, approved the final draft.
- Wei Yang conceived and designed the experiments, analyzed the data, prepared figures and/or tables, authored or reviewed drafts of the paper, approved the final draft.
- Chun-Ping Yang and Zhi-Yong Wang conceived and designed the experiments, approved the final draft.
- Jing Zhang and Hao Liu performed the experiments, approved the final draft.
- Li-Zhi Qian contributed reagents/materials/analysis tools, approved the final draft.

### Data Availability

Models and uncropped blots are available in Figs. 1–5 and raw data are available in the Supplemental Files.

### Supplemental Information

Supplemental information for this article can be found online at http://dx.doi.org/10.7717/peerj.7818#supplemental-information.

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
