# Peer review of "Bioinformatic and biochemical analysis of the key binding sites of the pheromone binding protein of Cyrtotrachelus buqueti Guerin-Meneville (Coleoptera: Curculionidea)"

_PeerJ, doi:10.7717/peerj.7818_

## Round 0.1 · original submission · Major Revisions

I have received three reviews for your manuscript and, based upon those reviews, feel it is likely to be suitable for publication if you can address a number of issues. Therefore my decision is for major revisions.

I'll start with a positive note. I feel that the title could be somewhat more powerful. Your protein expression and site-directed mutagenesis work adds a nice angle to this manuscript that is not really captured by the current title. I suggest that you change the title to reflect the fact that you have done some experimental work. A reasonable title might be:

"Bioinformatic and biochemical analysis of the key binding sites of the pheromone binding protein of Cyrtotrachelus buqueti Guerin-Meneville (Coleoptera: Curculionidea)"

Although I think biochemistry is rather limited, I think it is fair to let readers know that you did more than in silico analyses.

I'll start with a quick wave of the hand to reviewer #2, who offered a short review. All I'll say it that the small number of comments that were offered were reasonable; pay attention to them and address them.

Now I'll move on to the meatier reviews. Regarding reviewer #1, I am not bothered by the absence of physiological or behavioral assay. We have to start somewhere, and the molecular characterization you did is useful. I believe all of the other comments from reviewer #1 are valid and should be addressed. I would especially stress making sure you have included as much important literature as possible. If anything, I'd rather you err on the side of citing too many papers than on the side of citing too few.

I also think you should pay careful attention to reviewer #3 and address all of their comments. However, I would like to make some concrete suggestions regarding the suggestion that you make a phylogeny. I agree completely and was left wondering about several issues:

1. Are there any potential templates that exhibit more similarity to CbuqPBP1. I actually did a little poking around in pdb and was unable to find any, so I suspect you are using appropriate templates. However, I believe that you have to document this.

Along these lines, make sure you report enough information to address reviewer #1 questions about the quality of the model. I am actually fairly confident in your model, but I want to see documentation!

2. Is His53 conserved? It looks like it aligns with a Tyr in the aphid (NribOBP3) protein you used as a template, but I wonder if it is conserved in coleoptera. Or other insects. The same for Phe69 (which aligns with a Leu in Nrib). I think you can address these questions in the following ways:

a. Communication: Label the important residues in Fig. 1A. I had to look at Fig. 2C, figure out the sequence around His53, and then look for it in the pairwise alignment. Help your reader find the important residues! Some sort of indication of the residues, like an arrow with "His53" and "Phe69" is critical.

b. Evolutionary/Phylogenetic analyses. There are 21 available coleopteran genomes. I do not know how well annotated all of them are, but I know some are quite well annotated. There is no reason you couldn't build an alignment using selected coleopterans (and other holometabolous insects). Is the aphid sequence you used as a template an ortholog or paralog? There is one way to see - build a large alignment and pull together a tree.

c. Regarding tree inference, the sequences are short and fairly divergent (at least for Cbuq and Nrib). So I don't necessarily expect strong support. However, it would be perfectly reasonable to build an alignment using a standard tool like Mafft or Muscle and running an analysis with a good likelihood program like IQ-TREE. There are servers for the aligners and for IQ-TREE (IQ-TREE can also identify the best-fitting model of protein evolution, so the analysis should be straightforward). I would like to have at least some idea of the evolutionary relationships for the proteins you are examining.

Please comment on whether the important binding site residues are conserved within coleoptera (and any other groups). Also, if there appear to be a lot of gene duplications are they conserved within certain gene clades?

Also, please make sure to provide machine readable files with the sequence alignments and trees in addition to any figures (which could be supporting figures). By this I mean relaxed phylip or nexus format alignments and newick or nexus format trees (i.e., alignments readable by IQ-TREE and trees readable by a program like FigTree).

I hope you will be able to address these concerns and resubmit the manuscript.

·

Basic reporting

literature needs to be updated and relevant literatures are missing

Experimental design

some positive control for mutation experiment is useful, see my comments in comments fro authors

Validity of the findings

no physiology or behavioral validation

Additional comments

Yang et al., Bioinformatics analysis of the key binding sites of the pheromone binding protein of Cyrtotrachelus buqueti Guerin-Meneville (Coleoptera: Curculionidea).

Using Homology modeling and molecular docking they investigated to study how pheromone binding protein CbuqPBP1 and its ligand, dibutyl phthalate/cedar camphor interact. They have identified the binding sites; its 3D structure and they also validate the binding sites by doing site-directed mutagenesis. Which is interesting study.

The main shortcoming of the study is why they selected CbuqPBP1, and also where is the active site, in which of the 6 helixes?

We do agree that the authors did some quality check of the built 3D model. They used PROCHECK tool for the quality check. But RSMD is not presented. And only one tool may not be enough to assess the rationality of the model. The authors can add another tool such as 3D profile. What is the quality (Stereochemistry, resolution) of the template?

Alignment and homology modeling

Conventional Protocols for Homology modeling were followed.
template met criteria
3D structure built; quality checked
model optimized using robust web online server.

But authors did not give PDB accession number of the template and its Resolution. Given that the quality of the template is also based on its resolution. it is important to give the resolution of the initial template.

In the site directed mutagenesis we also need a positive control, to make sure other PBPs are not affected by the mutation.

What is the behavioral significance of the CbuqPBP1 and dibutyl phthalate interaction?

Is CH-π the binding site not well described it is not clear, where on the 3D structures, how its structure a little bit of description is needed.

What is the result implication we need some discussion about its significance?

Where are these amino acids on which helixes? Phe69 and His53

The objective was to study CbuqPBP1 - dibutyl phthalate interaction, but the result is talking about CbuqPBP1- cedar camphor interaction? Which one is the ligand and how it is selected?

Is cedar camphor correct name of the ligand?

The author wrote: Thus, His53 of CbuqPBP1 appears to be a key binding site residue, and Phe69 is a very important binding site for particular ligand interactions. Which particular ligand? You have to specify

Line 23. OBPS are not chemoreceptors?

Line 24 -25 give references

The English need to be revised for instance Line 25-27 not clear
Line 30-33 rephrase for instance …CH-π interaction between the benzene..., interaction is not
correct replace with interacts

Line 31 is CH-π is an active site of CbuqPBP1?

Line 35…. to which ligand?

Line 36 your objective and result are different

The introduction: shorten it by reviewing up to date knowledge of the subject matter (PBPS, your insect model and interaction between the protein and the ligand and possible implication) and the research gap you are addressing.

For instance, poor description of Pheromone binding protein

The authors have not stated how important is Pheromone communication in Cyrtotrachelus buqueti

the economic importance of the pest better way

Line 46, what do you mean by tentacles, do you mean sensilla or antenna? Tentacles is not the olfactory organ of insects

The authors need to update themselves about the biology of the olfactory system

what do you mean receptors are widely distributed? Do you want to say they are equipped with…?
Line 50 antennal binding protein, what do you mean? You need to use recent works about OBPs as reference

Line 115-116. Why did the authors choose this coordinate for the docking? No reference or justification

Line 125 we need more description of the method for instance, add the PCR conditions for any duplicability of the experiment

The study ends at bioinformatics data alone, no validation using physiology or behavior or interpretation. Nothing has been said about the implications of the study and how it will help to manage the pest.

Reviewer 2 ·

Basic reporting

There were many cases in the manuscript where the sentences including lines 80-82 were not properly structured, suggesting that the paper would improve if it could be edited for language and technical correctness. Every abbreviation for the Latin name of the species should be checked include lines 67,71, which should be accorded to the abbreviation rules.

Experimental design

no comment

Validity of the findings

no comment

Reviewer 3 ·

Basic reporting

In this study, the spatial structure of Cyrtotrachelus buqueti PBP1 was studied by constructing three-dimensional homoologous model, and the binding capacity of CbuqPBP1, cbuqpbp1-his53a, cbuqpbp1-phe69a mutants and Dibutyl phthalate, benzothiazole and cedar camphor were analyzed and compared by fluorescence competitive binding and molecular docking. The results showed that there were two important sites of CbuqPBP1, His53A and Phe69A.
Minor revision
(1) Line 180, CbuqPBO 1, should be CbuqPBP1
(2) Line 238-242, When the final concentration of odor substance was higher than 50 μM, the fluorescence peak did not decrease to half its value. This showed that almost no affinity was established between protein and the odor substance, and the binding constant could not be calculated (Fig. 6).
(3) In figure 4, Lane 3.4.5.6 was not marked in the diagram.

Experimental design

Major revision
(1) CbuqPBP1 is recognized as PBP protein, suggest author add the sequence alignment results with other insects PBP1 and evolutionary tree analysis.
(2) In Molecular docking results, why NribOBP3 protein was selected as model protein control? Maybe choose other insect PBP1 proteins that have resolved structures as models. The sequence similarity is probably higher than NribOBP3 protein.

Validity of the findings

Three kinds of fluorescent competitive binding substances used by CbuqPBP1 have been studied, and the site-directed mutagenesis have also been studied in PBP of other insect species, this experimental results in this paper also explain some olfactory molecular mechanism. It can also provide reference for other insect related research.

Additional comments

In this study, the spatial structure of Cyrtotrachelus buqueti PBP1 was studied by constructing three-dimensional homoologous model, and the binding capacity of CbuqPBP1, cbuqpbp1-his53a, cbuqpbp1-phe69a mutants and Dibutyl phthalate, benzothiazole and cedar camphor were analyzed and compared by fluorescence competitive binding and molecular docking. The results showed that there were two important sites of CbuqPBP1, His53A and Phe69A.
Minor revision
(1) Line 180, CbuqPBO 1, should be CbuqPBP1
(2) Line 238-242, When the final concentration of odor substance was higher than 50 μM, the fluorescence peak did not decrease to half its value. This showed that almost no affinity was established between protein and the odor substance, and the binding constant could not be calculated (Fig. 6).
(3) In figure 4, Lane 3.4.5.6 was not marked in the diagram.

Major revision
(1) CbuqPBP1 is recognized as PBP protein, suggest author add the sequence alignment results with other insects PBP1 and evolutionary tree analysis.
(2) In Molecular docking results, why NribOBP3 protein was selected as model protein control? Maybe choose other insect PBP1 proteins that have resolved structures as models. The sequence similarity is probably higher than NribOBP3 protein.

---

## Round 0.2 · Minor Revisions

Both reviewers feel that you have addressed all of their concerns regarding the manuscript and I concur. However, reviewer 1 also made that comment that there were numerous problems with the language and it could really profit from a detailed reading and revision by a native English speaker. I also concur with that assessment.

I am assigning this manuscript a decision of "minor revisions" because I feel the scientific content is adequate for publication but the language is not.

Obviously, if you have a colleague that is a native (or very fluent) English speaker that can help it would be good.

Once you have returned a version of the manuscript that adequately addresses the language issue it will be accepted.

·

Basic reporting

The authors have addressed the concern I have raised in my first review and the manuscript reads well. Deserve publication

Experimental design

addrresed

Validity of the findings

They used a different approach to prove their claim

Reviewer 2 ·

Basic reporting

This MS has been revised and it looks better for a publication now. However, after reading through the MS several times, I found some language problems which made me feel that some parts of the MS are weak or even a little poor in logic. For example, through the MS the authors used 11 times of “According to”, even from line 54 to 60, there are 3 times using of it in just 3 sentences, and this kind of using words made the part weird and, very poor in logic in my opinion. Besides, some other parts are similar as this one. Also, there are some grammatical errors, such as line 69 “The larva, in particular, prefer the…” should be “The Larvae...." Moreover, some words are used improperly and they are too many to be listed through the MS.
Furthermore, in the part of DISCUSSION, there are two independent paragraphs, line 258 to 263 and line 264 to 271, and could the authors integrate them into one paragraph? If not, please show the reasons.

Experimental design

no comment

Validity of the findings

no comment

Additional comments

I strongly suggested the MS should get language revise by English native speakers, and after the re-editing, the MS could be accepted.

---

## Round 0.3 · Minor Revisions

I would like to begin this decision with a brief apology. I was delayed in my response due to an unexpected family problem (a serious illness). I feel bad that it took me longer than I would have liked and feel I owe you an apology for the delay.

With my apology out of the way, I would like to accept this manuscript but I would like you to make three very minor modifications. First, one added sentence was very difficult to read, to the point I cannot understand exactly what it is saying:

On lines 85-88 in the reviewing pdf you state: "According to the system evolutionary tree,CbuqPBP1 was quite similar to PBP of other insects.Amino acid sequence similarity analysis showed that CbuqPBP1 had 37.68% similarity with 27 PBPs of 17 insects of Coleoptera and Lepidoptera. The similarities with Coleoptera and Lepidoptera were 38.47% and 52.39% respectively(Yang et al., 2018)."

The confusing point for me is that a single protein is extremely unlikely to have 37.68% similarity (or any other specific number) to 27 other proteins if those 27 proteins differ from each other. In fact, you state that the similarities to Coleoptera and Lepidoptera are different. This makes me suspect that you mean to say that the minimum similarity between CbuqPBP1 and any of the 27 PBPs is 37.68%. However, I am uncertain how that can be true if the similarity values for Coleoptera and Lepidoptera that you list are correct. I would really like to see this clarified.

Second, I recognize that, based upon your rebuttal letter, that a phylogenetic tree has been through peer review and has been published in Yang, H., T. Su, W. Yang, C.P. Yang, X.L.Zhou and Y. P. Li 2018. Molecular Cloning and Expression Analysis of CbuqPBP1 Gene in the Bamboo Snout Beetle, Cyrtotrachelus buqueti. Journal of Sichuan Agricultural University 36:78-85. However, I was unable to access papers from that journal and I suspect many international readers will also lack access. This creates a problem: I would really like to readers of PeerJ, which is open access, to have access to some minimum amount of information about the phylogeny. I think a reasonable compromise would be for you to make a very simple figure *based on* your published phylogenetic tree and put it in supporting information. Obviously, you don't want to use the same exact figure since that is part of a published paper. However, I don't think a very minimal figure that provides the same basic information is inappropriate or too difficult for you to produce. I think you can make a figure legend that simply states that the supporting figure is based on the phylogenetic tree from Yang et al. (2018). This figure could be added to the supplemental information and

Finally, the supplemental file C-CbuqPBP1align-best-jieduan.pse appears to be a photoshop elements file. This is not a very standard file format. If it really is a photoshop elements file it should be very easy to convert to a standard and easy to open file format, like tif, png, or pdf. Please do so.

Assuming my interpretation of the sentence is correct I suggest rewriting as follows:

"Based on a phylogenetic analysis (Yang et al., 2018; summarized in supplementary information) CbuqPBP1 was quite similar to the PBPs of other insects. The minimum amino acid sequence similarity to any one of 27 PBPs from 17 insects in Coleoptera and Lepidoptera. was XX.XX%. The minimum similarities with PBPs from Coleoptera and Lepidoptera were XX.XX% and XX.XX% respectively (Yang et al., 2018)."

Obviously, if I have misinterpreted your original sentence you should change this to more clearly explain the result.

I apologize for having to add an additional step, but I think it is important to provide this information to readers. I will try to help finalize the resubmission absolutely as quickly as possible once you do this.

---

## Round 0.4 · accepted · Accept

I appreciate your willingness to expand the supporting material to provide me (and other readers) with sufficient information regarding the material that was difficult for me to access (since I feel it will be difficult for at least some other readers to access this material)

Therefore I am happy to accept your manuscript. Please try to alert potential readers to the manuscript using social media - there are studies showing that papers publicized by social media are cited more than papers that are not publicized.